# Position: Token Taxes Can Mitigate AI's Economic Risks

Lucas Irwin [1]   Tung-Yu Wu [1]   Fazl Barez [1]

## Abstract

AI-driven automation threatens to erode government tax bases, lower living standards, and disempower citizens—risks that mirror the 40-year stagnation of wages during the first industrial revolution. While AI safety research has focused primarily on capability risks, comparatively little work has studied how to mitigate the economic risks of AI. This position paper argues that technical governance researchers should prioritize the study of token taxes: usage-based surcharges on model inference applied at the point of sale. We situate token taxes within previous proposals for robot taxes and identify two key advantages: they are enforceable through existing compute governance infrastructure, and they capture value where AI is used rather than where models are hosted. We then present a research roadmap. For enforcement, we outline a staged audit pipeline—black-box token verification, norm-based tax rates, and white-box audits—and identify open technical problems at each stage. For impact, we highlight the need for economic modeling of cost pass-through and deadweight loss. Finally, we discuss why FLOP taxes may be preferable, token taxes could stifle innovation, and how to prevent AI superpowers from vetoing such measures.

## 1. Introduction

**Position: This position paper argues that the ML research community should prioritize the study of token taxes as a technical governance research agenda. Recent work in AI safety and economics has demonstrated that AI could pose serious economic risks to society, ranging from high unemployment to fiscal crises to the gradual disempowerment of citizens (Kulveit et al., 2025; Hampole et al., 2025; Frank et al., 2025). We claim that token**

---

[1]Department of Engineering Science, University of Oxford, Parks Road, Oxford, England, United Kingdom, OX1 3PJ. Correspondence to: Lucas Irwin <lucas.irwin@stcatz.ox.ac.uk>.

*Proceedings of the $43^{rd}$ International Conference on Machine Learning*, Seoul, South Korea. PMLR 306, 2026. Copyright 2026 by the author(s).

**taxes are well placed to mitigate AI's economic risks due to their unique advantages in being enforceable and usage-based.**

Throughout history, technological revolutions have displaced workers before eventually ushering in new forms of work. During the industrial revolution, workers ultimately found new work in the mechanized factories which replaced them (Allen, 2009a). However, societal adjustment is often preceded by technology-induced upheaval. During the first industrial revolution, a period now referred to as *Engel's Pause* (1790-1830) saw real wages stagnate as GDP per worker increased rapidly (Allen, 2009b; Frey, 2019; 2025). Engel's Pause triggered a downturn in living standards, as children worked in factories to support their parents and diseases such as cholera saw a substantial uptick in cases (Allen, 2009b).

The prevailing view within the AI research community asserts that AI will create new jobs, increase wages, and boost productivity (Butler et al., 2023; Brynjolfsson et al., 2018; Hampole et al., 2025). This has led the AI safety community to focus heavily on capability threats and neglect economic risks. However, the historical example of Engel's Pause shows us that, while new jobs may be created in the long term, living standards may still plummet for the average worker for 40 years. Early evidence of AI's impact on unemployment is already emerging in studies which show unemployment in AI-exposed early career roles rising by 16% while levels for experienced workers remain constant (Brynjolfsson et al., 2025).

Indeed, a recent report from the White House underscored the economic risks posed by rapid AI development (The White House, 2026). In it, the authors state that estimates surrounding AI's impact on global GDP range from 1% to 45%. The report argues that low estimates are now unlikely since AI investment already boosted US GDP by 1.3% in the first half of 2025 alone - a rate rivaling the scale of railroad investment in Britain during the first industrial revolution (Pereira et al., 2014). If AI develops agency, or rapidly boosts worker productivity without creating new labour demand at the same rate, unemployment could rise as government labour income decreases (Ayres, 1990; Donaldson, 2018).

Prior work has argued in favour of *robot taxes* as a solution

to the increasing economic inefficiencies created by technology and automation (Abbott & Bogenschneider, 2018; Mazur, 2018; Cabrales et al., 2020). Proponents argue that labour tax rates are much higher than capital tax rates because governments seek to incentivize innovation. Yet this system ceases to be efficient when capital **becomes** the labour. In the case of AI, foundation models are treated as capital even though they carry out tasks usually performed by human labour (Acemoglu et al., 2020). A robot tax seeks to solve this problem by restoring *tax neutrality* to the system. This is widely argued to be efficient since robot taxes help balance the scales by removing incentives for companies to choose AI workers over human workers (Abbott & Bogenschneider, 2018; Mirrlees et al., 2011). Prior suggestions for robot taxes include raising the corporate tax rate, levying an "automation tax", and disallowing corporate tax deductions whenever a company invests in automation (Abbott & Bogenschneider, 2018).

This position paper argues in favour of prioritizing the *token tax* as a field of technical governance research. Defined as a usage-based surcharge on model inference applied at the point of sale, token taxes are unique in two key respects. First, they are (1) enforceable: existing compute governance infrastructure enables token taxes to be audited and enforced using black-box or white-box methods, adding an extra technical layer to existing tax auditing methods (Sastry et al., 2024). Second, they are (2) usage-based rather than firmbased. This enables taxation at the point of sale, which means the tax mitigates inequality by capturing value where the AI is used rather than where the model is hosted.

We discuss the economic risks of AI, including (1) government fiscal crises, (2) gradual disempowerment (3) lower living standards, and (4) global inequality. We then situate the token tax within previous proposals for robot taxes and present the following research roadmap: (1) black-box technical auditing, (2) norm tax rates for model categories, (3) agent-based modeling of economic impacts, (4) solving the self-hosted model problem, and (5) a comparative analysis of mitigation strategies.

Finally, we consider the following alternative viewpoints: (1) token taxes will disincentivize innovation, (2) token taxes do not account for Jevons' Paradox, (3) AI superpowers can veto token taxes, and (4) an FLOP tax is preferable to a token tax. Despite the many uncertainties surrounding token taxes, we argue that the evidence points towards the tax being a promising technical governance research area for mitigating the economic risks of AI.

## 2. Risks

Thus far, AI safety has focused heavily on the risks of increasing AI capabilities (Bostrom, 2014). Risks pertaining to the future of work have been less prominent but may be even more urgent than capability risks (Acemoglu & Restrepo, 2019). We discuss four key economic risks of AI, motivating our argument in favour of promoting token taxes as a field of technical AI governance. For clarity, we do not assume that severe economic risks such as mass technological unemployment are inevitable. Rather, we sketch out possible economic risks that would result in societal destabilization over time.

### 2.1. Government fiscal crises

The first economic risk of AI we consider is the loss of government revenue, which would result in fiscal crises. Early studies have shown that high exposure to AI in companies increases unemployment risks while reducing labour demand (Hampole et al., 2025; Frank et al., 2025). Recent work has also found that as automation increases, government fiscal revenues decrease, because taxes on labour are the highest source of government revenue (Casas & Torres, 2024). High unemployment would also reduce consumption and increase states' fiscal costs, eroding the two main sources of public finance (Korinek & Lockwood, 2025). While the jury is still out on whether AI will cause mass unemployment, existing empirical evidence points to the risk of fiscal crises.

Furthermore, since frontier AI companies are multi-national corporations (MNCs), they are able to effectively evade existing taxes by transferring ownership of their intangible, digital assets to affiliate corporations in low-tax jurisdictions (Lowry, 2019). Existing digital services MNCs avoid tax by recording profits in more favourable tax jurisdictions and engaging in strategies such as debt and earnings stripping (Gravelle, 2009). This results in a loss of tax revenue for the state in which the corporation serves customers - a phenomenon which, until recently, remained unaccounted for by international law regimes (Lowry, 2019; Cui, 2019). Tax avoidance would therefore further exacerbate AI-induced government fiscal crises.

### 2.2. Engel's Pause: Lower living standards and stagnating wages

The second major risk is an offshoot of the first: if governments lose income and workers lose their jobs, living standards will plummet. The prevailing view within the AI research community is that AI is a general-purpose technology that will create new jobs and increase productivity (Butler et al., 2023). This rhetoric reflects the idea of the *productivity bandwagon* coined by Nobel prize-winning Economist Daron Acemoglu and Simon Johnson (Johnson & Acemoglu, 2023). The productivity bandwagon states that technology will automatically advance societal progress by increasing productivity. However, Acemoglu and Johnson assert that technological progress only produces broad

prosperity if it (1) increases worker marginal productivity and (2) workers have sufficient bargaining power to claim a share of the productivity gains (Johnson & Acemoglu, 2023).

When power looms displaced hand-weavers during Engel's Pause, productivity increased, but wages stagnated and working conditions worsened because workers did not have the requisite bargaining power (Allen, 2009b). Examples such as these illustrate the danger of relying on the productivity bandwagon to distribute gains. The recent White House report's acknowledgment of the similarity of the AI revolution to the scale of the industrial revolution underscores the risk of Engel's Pause recurring in the 21st century. AI progress therefore threatens to impose similarly deleterious effects on average living standards.

### 2.3. Gradual disempowerment of citizens

Thirdly, we consider the *gradual disempowerment* of citizens (Kulveit et al., 2025). If AI displaces humans by out-competing them in nearly all economic and societal functions, humans will gradually lose control over the levers of state power. Currently, governments and economic systems respond to human action since humans are the main drivers of economic and social progress (Giddens, 1984). If AI begins to displace humans as the main driver of economic growth, governments will become far less responsive to citizens' needs, since they will no longer need their support (Kulveit et al., 2025).

A similar phenomenon occurs in so-called *rentier states* (Neal, 2019). States such as Venezuela, Saudi Arabia and Oman have abundant resources and earn most of their income from oil rents as opposed to their citizens' labour. Yet most of their citizens live in abject poverty. This is referred to as the *resource curse* (Acemoglu et al., 2002; 2005). A core reason for the resource curse is the fact that states no longer have an incentive to look after their own people. In the context of AI-driven unemployment, states will make most of their revenue from taxing AI capital and not workers and hence lose their incentive to tend to citizens' needs. We argue that it is imperative that robot tax legislation such as the token tax is passed preemptively before governments lose all incentives to do so.

### 2.4. Global inequality

Finally, we consider the risk that AI could exacerbate global inequality. Absent new forms of taxation, AI's economic benefits are likely to be unevenly distributed, worsening existing divides between rich and poor countries. This is obvious when analysing the global distribution of compute, models and agents.

*Compute:* The advanced compute chips necessary to train

and run frontier AI models are highly concentrated in a handful of economically developed nations. These nations, referred to as the "Compute North" in recent work, have priority access to and jurisdictional authority over AI chips, while countries in the "Compute South" must rely on renting these chips from the Compute North (Lehdonvirta et al., 2024).

*Models:* The vast majority of frontier model companies are also highly concentrated in a handful of rich countries. OpenAI, Anthropic, Google DeepMind and xAI are US-owned while DeepSeek and Alibaba are Chinese-owned. Second-rate model providers such as Mistral (France) and G42 (UAE) are also headquartered in wealthy countries contained within the "Compute North". These countries will therefore benefit greatly from their ability to determine tax rates on these resources, while countries outside of the Compute North will be left behind. Given that the Compute North and South already reflect existing global inequalities in living standards, AI is likely to exacerbate this divide unless new usage-based taxation systems which benefit both the Compute North and South are developed.

*Agents:* The capabilities of advanced agents are also likely to vary greatly from country to country. The concept of *agentic inequality* refers to the disparities in access to and capabilities of agents across three dimensions: availability, quality, and quantity (Sharp et al., 2025). Disparities in access could reduce the social mobility of people lacking access to the best agents, worsening the economic gap between the Global North and South.

## 3. The Token Tax

To mitigate AI's economic risks, we argue that the technical AI governance community should prioritize research on token taxes. We situate token taxes within existing research on robot taxes, present the unique advantages of token taxes, and outline a 3-stage audit pipeline for enforcement.

### 3.1. Economic background:

Recent work by Korinek & Lockwood (2025) distinguishes between two stages of AI-driven economic transformation which influence the design of a token tax (Korinek & Lockwood, 2025).

**Stage 1: The post-labour economy** At this stage, labour's share of income declines significantly but humans remain the primary consumers of resources. Tax policy must therefore adapt to maintain government revenue.

**Stage 2: The AGI economy:** During the second stage, AGI both produces and consumes economic resources, substituting human labour entirely. Tax policy must therefore focus on determining what percentage of AGI capital to "harvest".

As AI automation displaces human labour, the question of optimal taxation on AI resources becomes central. Korinek & Lockwood (2025) formally define the following economic model of labour tax revenue as a share of GDP.

Let $\alpha$ denote the capital share of income and $\varepsilon$ denote elasticity governing the taxable labor base. Then,

$$\max \frac{\text{labor tax revenue}}{GDP} = \frac{1 - \alpha}{1 + \varepsilon(1 - \alpha)}. \quad (1)$$

As labour's tax revenue share of GDP approaches 0, capital's share approaches 1. When $\alpha \to 1$, labour's share of income vanishes; for a full derivation, see Section A.1.4 of Korinek and Lockwood (Korinek & Lockwood, 2025).

Aligning with the authors, we consider consumption-based taxes to be the primary instrument for keeping this ratio stable, and thereby mitigating AI's economic risks during stage 1 of AI-driven growth. In this position paper, we focus solely on stage 1 since stage 2 only occurs in the most extreme AI forecasts.

### 3.2. Robot Taxes

Prior work has argued in favour of robot taxes as a solution to the increasing economic inefficiencies created by technology and automation (Abbott & Bogenschneider, 2018; Mazur, 2018; Cabrales et al., 2020).

*Table 1.* Comparison of forms of robot taxes across three metrics: *usage-based*, *auditability*, and *innovation risk*.

| Tax type | Usage-based | Auditability | Innovation risk |
|---|---|---|---|
| Corporation tax | No | High | High |
| Automation levy | Yes | Low | Medium |
| Deduction disallowance | Yes | Low | Medium |
| Payroll tax reform | No | High | Medium |
| Capital gains / wealth tax | No | Medium | Medium |
| FLOP tax | Yes | High | High |
| **Token tax** | **Yes** | **High** | **Low** |

**Definition:** *A robot tax is a fiscal instrument designed to restore tax neutrality between human labour and automated labour. Robot taxes ensure that the tax rate on automated labour approximates the rate applied to human labour (Abbott & Bogenschneider, 2018; Mirrlees et al., 2011).*

The economic motivation behind robot taxes stems from the misalignment which results when automated labour is taxed as if it were capital. In general, labour tax rates are substantially higher than capital tax rates because governments seek to incentivize innovation. In the case of AI, foundation models are treated as capital even though they carry out tasks usually performed by human labour (Acemoglu et al., 2020). This creates an incentive for companies to choose automated labour over human labour for the same job. A robot tax seeks to correct this by removing the implicit fiscal

incentive for companies to substitute AI workers for human workers (Abbott & Bogenschneider, 2018; Mirrlees et al., 2011).

Existing proposals for robot taxes include (1) raising the corporate tax rate, (2) levying an automation tax triggered by automation-induced layoffs, (3) disallowing corporate tax deductions for capital investments in automation, (4) reforming payroll taxes to remove the fiscal bias against human labour, and (5) taxing the capital gains and wealth accruing to automation's beneficiaries (Abbott & Bogenschneider, 2018; Mazur, 2018; Cabrales et al., 2020).

Table 1 situates the token tax within this landscape of existing proposals. We also include FLOP (floating-point operation) taxes as an alternative, which are similar to token taxes but treat FLOPs as the unit of taxation instead of tokens. Unlike all other forms of robot tax, token taxes are unique in being both usage-based and auditable, while also minimizing innovation risk. We discuss the token tax's unique advantages in further detail in the following section.

## 4. Technical Implementation:

**Definition:** *A token tax is a usage-based surcharge applied to model tokens at the point of sale.*

**Tax collection:** Drawing on existing usage-based billing procedures for language models, the token tax would be levied as a percentage markup on the provider's billed token cost paid directly to the government. For instance, if the token tax is 10% and the cost-per-token for a given model is $1, the AI company will pay $0.10 in token tax. The tax will be collected as a consumption tax on tokens used during inference. Importantly, we also align with the authors by asserting that token taxes should only apply to final consumption and not intermediate consumption (a business-to-business (B2B) chatbot would not be taxed). By applying token taxes to final AI services, we avoid distorting AI development and thus minimize the disincentive to invest in AI infrastructure (Korinek & Lockwood, 2025).

**Tax rates:** Tokens are not a standard unit of measurement. Their count depends on (1) the tokenizer and (2) the language used. For the same sentence written by a specific model, the number of tokens decoded by tokenizers of different LLMs varies. For the same sentence written in two different languages, the number of tokens decoded by tokenizers of those languages also varies.

We do not argue in favour of a single, uniform tax, but rather for individual token tax rates for model families depending on (1) and (2). We are less concerned with (1), as the token tax will incentivize AI model providers to optimize tokenization and thus minimize energy usage. We do consider (2) a valid concern, which we address through our 3-stage audit

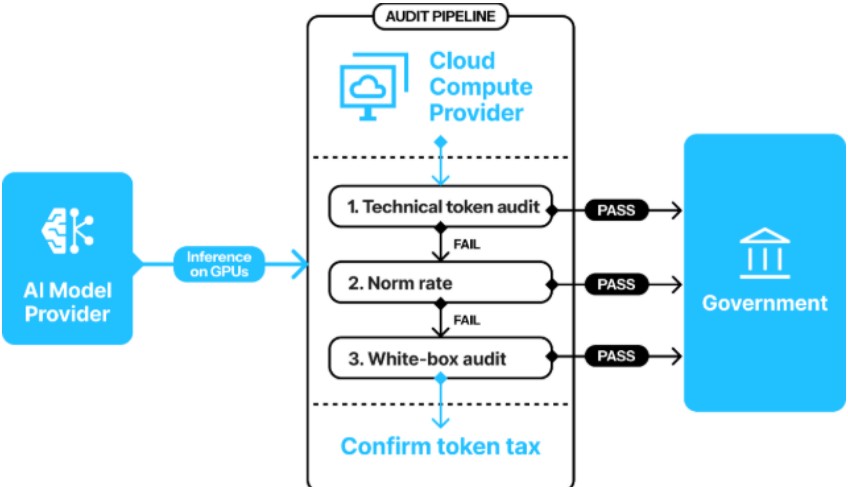

*Figure 1.* **Token tax collection procedure.** The cloud compute provider acts as an intermediary between the AI model provider and the government. It runs inference on behalf of the AI model provider, collects billed tokens, and determines tax liability through a staged audit pipeline. The provider first attempts (1) a black-box technical token audit. If this fails, it applies (2) a norm-based tax rate derived from average token usage. If the norm-based tax approach fails, it escalates to (3) a white-box audit. Once an audit pathway succeeds, the compute provider confirms the total token tax and reports it to the government.

pipeline.

**Advantages:** We argue that there are two main advantages of using a token tax over other forms of robot tax. First, token taxes could potentially be effectively enforced by governments using: (1) black-box token audits, (2) norm-based taxes, and (3) white-box audits. To facilitate effective enforcement, we argue that compute providers should act as intermediaries between the government and AI model providers to apply a staged audit pipeline of (1), (2), and (3) (see Figure 1). **N.B** We do *not* automatically assume that the compute provider and the AI model provider are the same entity. If this is the case, we argue that the same requirements described below would apply to the combined entity. Second, we claim that token taxes mitigate against the worsening of global inequality. By collecting the token tax where the tokens are utilized rather than where the model is hosted, token taxes ensure that AI consumer countries (such as those in the Compute South) also benefit from AI tax receipts.

### Advantage 1: Auditing & Enforcement

Many generative models take in tokens as inputs and return tokens as outputs during inference. However, since only model providers have full access to the generative process undertaken when a user calls an API, and users and auditors only have access to the outputs, companies can misreport the number of tokens used in generation (Velasco et al., 2025a; Bourrée et al., 2025; Velasco et al., 2025b). This enables providers to inflate their profits when using token-based pricing (Velasco et al., 2025a; Wang et al., 2025; Sun et al., 2025).

We argue that this problem can be resolved by requiring compute providers to implement the following 3-stage audit pipeline:

***Audit Stage 1: Black-box token audits*** The first audit stage requires compute providers to carry out black-box technical token audits. Compute providers serve as *record keepers*, *verifiers*, and *enforcers* and are therefore able to ensure AI companies comply with obligations imposed on them by governments (Heim et al., 2024). Delegating the task of auditing token count to compute providers can therefore solve the misreporting problem since providers can act as independent verifiers of compliance. Cloud hyperscalers already collect metadata on the compute consumed and the type of workloads (inference or pre-training) submitted to their servers (Heim et al., 2024). By mandating that providers also collect token-level usage data while preserving privacy, cloud providers can carry out oversight functions on behalf of the government. This would enable tax authorities to cross-check reported token usage against independent logs, barring AI labs from misreporting their usage to maximize profits.

***Audit Stage 2: Norm-based taxes*** The second audit stage draws on a common solution to tax evasion in rentier states such as Norway: *norm taxes* (Morgan & Robinson, 1976; Takle, 2021). The idea of a norm tax is simple: if a company decides to inflate its profits by *strategically under-reporting* token counts, auditors can refer to norm-based tax rates for each model category and charge the company according to the norm rate. This would require a methodology for continuously updating the average number of tokens used in every inference run for each type of model. It would

limit the amount that companies can inflate their profits by ensuring that a company must pay a flat rate based on empirical evidence.

It also has the added benefit of only requiring black-box access to models. This stage addresses the fact that different languages have variable tokenizations, since a methodology for calculating norm rates would also involve developing separate tax rates for different languages.

*Audit Stage 3: White-box audits* The third audit stage would require companies to share information about the generative process with third-party auditors. If companies are required to share this by law, they will no longer be able to game the number of tokens they report since auditors will have access to the information necessary to confirm the token count. The downside of this option is that it requires "white-box" access to models, and thus should be used as a backstop.

This three-stage audit pipeline ensures that token taxes are enforceable, in contrast to other forms of robot tax. Currently, white-box audits would be required since black-box token audits and norm taxes are unsolved problems. This motivates the technical governance research agendas we outline below.

**Advantage 2: Usage-based taxation**

In addition to being enforceable, token taxes also have the advantage of being usage-based. Usage-based token taxes would allow countries served by AI model providers to also benefit, preventing the worsening of economic divides between the global North and South. To prevent an unintended suppression of domestic small-medium enterprises (SMEs), the token tax would also only apply to AI companies making more than a set threshold of revenue.

*Token Tax vs Robot taxes:* The usage-based nature of token taxes also distinguishes the tax from other forms of robot tax. Tokens are billed to a user in a particular jurisdiction (Greece/Argentina etc). By choosing to tax automation or raising the corporate tax rate instead of taxing tokens, AI companies may simply choose to run inference on data centers in low tax jurisdictions to minimize their tax burden. With a token tax, companies must pay tax at the point of use, since the tax will be tied to the tokens billed to the user. This ensures that countries which do not have their own foundation model companies also benefit from robot taxes.

## 5. Our Recommendations

We believe the potential advantages of a token tax are compelling, but several technical governance challenges associated with implementing the tax must first be solved to pave the way towards feasibility. Here, we outline five recommendations for technical governance research agendas.

**R1: Black-box auditing**

In order to reliably audit token taxes, auditors must have effective technical methods at their disposal to verify token usage. Prior work has proposed methods for auditing tokens in language models and serves as a firm grounding for future research (Casper et al., 2024; Cai et al., 2025; Yuan et al., 2025). However, recent work has also demonstrated that significant challenges exist when verifying token counts with only black-box access to models (Velasco et al., 2025a;a; Bourrée et al., 2025). Since only model providers have white-box access to models, the asymmetry in information can be exploited to misreport token usage.

**Threat model:** Our adversary is a tax-evading, profit-maximizing AI company. The AI company possesses white-box access to model architecture, tokenizers, and weights. This allows the adversary to *strategically under-report token counts* in API calls by generating multiple valid tokenizations of the same text. It also enables the adversary to *secretly substitute cheaper models* for more expensive models as in the "model substitution problem" (Cai et al., 2025), and *use hidden reasoning tokens* which are not visible to the auditor (Sun et al., 2025). The adversary is incentivized to minimize their token tax bill by overcharging users while misreporting token counts to authorities. On the other hand, auditors only have black-box access to models via access to cloud provider logs. The limitations associated with black-box access are well documented in prior work (Casper et al., 2024), motivating our recommendations for technical governance solutions to black-box token verification.

**Technical governance solutions:** We recommend that researchers focus on the following areas of research towards reliable black-box audits of token counts. First, building on recent work defining black-box watermarking methods for mitigating harms in language models (Kirchenbauer et al., 2023), researchers should develop *cryptographic watermarking* methods to verify token counts and model type. Core research questions include whether watermarking can provide certifiable guarantees on token count, reliably detect the type of model used, and remain robust against adversarial tokenization. Second, regulators will need to reliably *audit hidden reasoning tokens* which count towards token usage but are not billed to the user directly. Researchers should therefore build on previous work to develop black-box verification methods for estimating hidden reasoning token counts from question-answer pairs without access to reasoning traces (Wang et al., 2025). Third, researchers should build on recent trusted execution environment (TEE) approaches for inference time auditing (Schnabl et al., 2025; Heim et al., 2024) to build auditing methods for token counts. Extending TEEs to token auditing could allow auditors to confirm token counts directly at inference time. Key research questions include preserving confidentiality

while enabling third party verification, minimizing inference overhead, and scaling.

We recommend that the technical AI governance community draw on this existing research to develop effective technical methods for auditing tokens. This recommendation is especially relevant to enable black-box auditing of token counts and prevent the need for white-box auditing which is much harder to implement in legislation due to intellectual property concerns (Casper et al., 2024).

### R2: Norm Taxes

Our second recommendation to the technical governance research community is to focus on the development of a methodology for defining *norm tax rates* for each model category. If a regulator is unable to reliably verify token counts, they should be able to instead refer to a regularly updated database of norm taxes for each model. Regulators would then only need to confirm the type of model used, and refer to the norm tax rate for an average inference run.

**Technical governance solutions:** We recommend that technical governance researchers devote time towards developing norm taxes by drawing on the *model substitution problem* (Shao et al., 2025). By developing effective auditing methods to verify model type, an independent commission could refer to data on average inference runs for each model and charge the model provider the norm rate if the reported token use falls below an empirically motivated threshold. Methods for watermarking, and TEEs suggested in R1 can be applied here but should also be paired with research on calculating average usage patterns for AI models. We recommend that technical governance researchers *partner with AI model companies* such as Anthropic and Perplexity to leverage their data on AI adoption and usage patterns to continually update norm rates (Appel et al., 2025; Yang et al., 2025). Relevant research questions include designing an independent commission responsible for norm tax rates, determining the optimal schedule for updating norm rates, and developing international cooperation strategies.

### R3: Agent-based modeling of economic impacts

Thirdly, we recommend that technical AI governance researchers partner with economists to study and model the impact of the token tax on the economic system. We discuss two crucial economic impacts to model. (1) *Cost pass-through:* The token tax will increase costs for AI providers. Higher costs for companies that adopt AI to replace human workers may be passed down to general consumers, as is the case with tariffs. In the worst-case scenario, the increased price of AI may squeeze out users who pay at the borderline of their budgets, aggravating the digital divide. (2) *Deadweight loss:* Levying taxes on a product shifts the supply-demand equilibrium, inevitably diminishing both the consumer and producer surplus. Although the govern-

ment captures a portion of this welfare as fiscal revenue, the process inherently results in a permanent loss of social welfare, known as deadweight loss, due to suppressed market activity.

**Technical governance solutions:** To estimate the economic effects of token taxes, technical governance researchers should partner with economists and *leverage Agent-Based Modeling (ABM)* to predict the impact of token taxes on markets. ABMs have already been successfully applied to make predictions in economics (Axtell & Farmer, 2025; Vu et al., 2022), with LLM-based ABMs already simulating large-scale social interactions in digital environments such as Twitter and Reddit (Yang et al., 2024). We recommend that the technical governance research community partner with economists to estimate the economic impacts of token taxes on real-world markets under different hypothetical AI growth scenarios. Relevant research questions include whether AI-intensive firms are sensitive to fluctuations in API costs (cost pass-through), the elasticity of consumer demand for services where AI has supplanted human labor (such as automated customer support), and practical mechanisms for mitigating deadweight loss, anchored in established economic frameworks such as optimal taxation theory (Ramsey, 1927). The technical governance community can provide unique skills in this domain by understanding the policy significance of the work while contributing technical expertise.

### R4: Solving the problem of self-hosted models

A core issue faced by our auditing framework is the problem of self-hosted models. We therefore recommend that technical governance researchers address the issue by drawing on compute governance.

**Threat model:** Our adversary is a company attempting to evade token taxes by avoiding using APIs altogether and hosting their own internal open-source models, hidden from regulators. Unlike AI companies which can be audited via compute providers which serve as natural intermediaries (Heim et al., 2024), self-hosted models lack this enforcement mechanism.

**Technical governance solutions:** We recommend that the technical governance research community devote time towards developing auditing mechanisms for companies self-hosting their own models. Methods for watermarking, hidden reasoning token audits, and TEEs suggested in R1 can be applied here, but must also be paired with a method for verifying which companies currently use AI labour. Prior compute governance research provides a starting point for addressing self-hosted deployment. Sastry et al. (2024) propose that governments should maintain an international registry of advanced GPUs to track entities capable of training and running frontier models (Sastry et al., 2024). We there-

fore recommend that technical governance researchers focus on *hardware-level monitoring*. Relevant research questions include whether hardware-level attestation via TEEs can verify GPU usage patterns (Schnabl et al., 2025), feasibility studies of international GPU registries, and on-chip mechanisms for tracking advanced GPU chips. By building on existing methodologies for measuring and tracking the global distribution of public cloud compute availability. (Lehdonvirta et al., 2025), the ML community should develop auditing regimes to keep track of self-hosted models.

### R5: A comparative analysis of mitigation strategies

Token taxes are an important tool for mitigating the economic risks of AI. However, they are only one component of an overall mitigation strategy. We therefore recommend that technical governance researchers focus on comparing the efficacy of token taxes to other alternative solutions. Alternative robot taxes, a tax on floating-point operations (FLOPs), AI sovereign wealth funds, decentralized AI, and AI as a public utility could all supplement token taxes. The design of a token tax will have to be informed by balancing the relative importance of complementary solutions.

## 6. Alternative Views

While we argue that token taxes are a promising technical governance tool worthy of study, we recognize core counterarguments to our case which we present in detail below.

### A1: *Token taxes will disincentivize innovation*

A salient counterargument to token taxes is that they will discourage innovation and incentivize leading AI firms to relocate to more favorable tax jurisdictions. Substantial empirical literature provides evidence that high-level tax rates affect where frontier technology firms choose to headquarter themselves: Innovative firms are likely to move to countries with lower marginal tax rates (Akcigit et al., 2016). Higher personal and corporate tax rates reduce the number of patents, innovation output and mobility, especially in skill-intensive sectors such as the AI industry. In general, the public finance literature underscores the way taxes on highly elastic industries such as AI can generate significant efficiency losses (Mirrlees et al., 2011).

The evidence therefore poses considerable challenges for proposals which seek to increase the marginal cost of AI inference. However, optimal tax theory also emphasizes the importance of tax design on economic outcomes (Mirrlees et al., 2011). Threshold-based taxes, exemptions for SMEs, and a gradual phase-in only once AI profits reach transformative levels, could substantially mitigate the negative impact of taxes (Lowry, 2019). The use of agent-based modeling to predict the impact of token taxes on other important economic indicators such as employment rates, wages, and inequality (Axtell & Farmer, 2025), could also be leveraged to minimize the negative impact on innovation.

Importantly, Korinek & Lockwood (2025) argue that token taxes on final consumption constitute optimal tax policy for stage 1 economies. Unlike capital taxes, consumption taxes preserve the incentive to save and do not discourage investment in AI development. On this view, failing to tax consumption during stage labour displacement would in fact be *inefficient* - a conclusion independently drawn by economists in favour of robot taxes (Mazur, 2018; Abbott & Bogenschneider, 2018).

### A2: *Token taxes do not account for Jevons' paradox of induced demand*

A second objection to token taxes is that it fundamentally misunderstands *Jevons paradox*. In the 19th century, William Stanley Jevons asserted that improvements in steam engine efficiency made coal-powered energy cheaper and hence made it more widely used (Jevons, 2023; Alcott, 2005). Applied to AI, this view holds that the more we allow AI to develop unfettered, the cheaper it will become and the more people will use it. As a result, AI applications will diffuse broadly throughout the economy, generating productivity and welfare gains for all. By artificially inflating prices above efficient levels, a token tax would blunt the induced demand and prevent AI from achieving its full potential. On this view, token taxes would risk suppressing the widespread adoption that would allow AI to drive societal progress.

We respond that token taxes could be designed to apply only once we start to see transformative economic growth due to AI. By drawing on threshold-based taxation to tax firms once profits exceed an empirically motivated threshold, we can allow Jevons' Paradox to unfold. Additionally, we claim that the economic risks of AI - unemployment, Engel's Pause, global inequality - outweigh the benefit of lower prices even if the token tax does blunt Jevons' Paradox.

### A3: *AI superpowers can veto token taxes*

A further objection to token taxes is geopolitical: Since AI superpowers (the USA and China) contain the vast majority of the AI supply chain, they are able to leverage their economic and diplomatic power to veto or undermine token tax regimes proposed by smaller nations seeking to implement AI-related legislation (Emery-Xu et al., 2025). Contentious negotiations surrounding Digital Services Taxes (DSTs) applied in many European countries serve as a precedent. Due to the fact that many digital services MNCs operating in Europe are American, the United States threatened retaliatory tariffs against any country refusing to revoke its DST, demonstrating the leverage tech superpowers have over other nations (Devereux & Vella, 2018).

To mitigate these trade tensions and retaliation from AI superpowers, we argue that regional agreements by coalitions of the willing could be established. The EU's market size and regulatory capacity enabled the successful implementation of the GDPR and the AI Act despite U.S. opposition (Act, 2024). A coordinated token tax by a coalition of the willing would be much harder to veto than unilateral national measures.

**A4:** *A FLOP tax is preferable to a token tax*

A FLOP-based tax directly levied on compute represents one of the most serious alternatives to a token tax. Compute-based thresholds are already applied as regulatory proxies for model capability and risk in AI legislation such as the EU AI Act and Biden's executive order 14110 (Act, 2024; Joseph R. Biden, Jr., 2023). Both use training compute (measured in FLOPs) as a proxy for model capability and risk, and recent technical reports by the European Commission's Joint Research Centre explore how cumulative compute can be measured and verified for this purpose (Meltzer & Tielemans, 2022). Instead of taxing tokens at the point of sale, governments could implement a levy on computing.

While FLOP taxes may be easier to audit than token taxes, they are less flexible than token taxes. Inference occurs across diverse deployments, including fine-tuned and hosted models - information which is captured by tokens but not captured by FLOPs. By auditing tokens, we are able to provide a more granular, though technically challenging, basis for taxation by defining different tax rates and norm rates for models.

Furthermore, FLOPs are an unstable measure of the economic value displaced by AI. Imagine that a model provider, $M$, offers a model that generates code for a website with a fixed prompt. Initially, the model requires $F$ FLOPs to generate 1000 output tokens. Now assume that over time, $M$ makes optimizations to its model which enable it to generate the same 1000 output tokens with $\frac{F}{2}$ FLOPs. Those 1000 tokens displace the same magnitude of economic value, yet the FLOP tax paid would be half of what the company paid before. By contrast, the token tax paid would remain constant since the number of output tokens has remained unchanged. Thus, a token tax is a more stable proxy for the economic value displaced by AI than a FLOP tax.

We also argue that token taxes and FLOP taxes are not mutually exclusive. An optimal policy framework might combine token taxes with FLOP taxes in a hybrid approach which leverages the enforcement advantages of FLOP taxes while preserving the consumption-based and equity-promoting features of token taxes. Indeed, we welcome proposals to combine FLOP taxes with token taxes.

**A5:** *Establishing a token tax regime would be a bureaucratic nightmare*

A final counterargument to a token tax is that the infrastructure required to implement the tax will impose high bureaucratic costs on governments. The costs of establishing an independent commission as recommended in R2, implementing a 3-stage audit pipeline, and consistently updating norm tax rates will be significant. While the costs are likely to be significant, we assert that these costs would pale in comparison to the magnitude of lost government fiscal revenues we highlight in risk 2.1. Therefore, the benefits of implementing a token tax regime would vastly outweigh the bureaucratic costs of establishing it. We also advocate for eliminating any unnecessary bureaucracy to minimize cost.

## 7. Conclusion

In this position paper, we claim that technical governance researchers should prioritize the study of token taxes. In contrast to existing robot taxes, we argue that the token tax offers 2 unique advantages: it is enforceable via existing compute governance infrastructure and it reduces global inequality by capturing value where AI tokens are used not where models are hosted. While uncertainties surrounding the token tax's economic impact and technical feasibility exist, we believe the evidence in favor of token taxes and the benefits we outline here justify further research into the five technical governance research agendas we have identified.

## Impact Statement

Our argument emphasizes the underappreciated economic risks of AI and advocates for a technical governance research agenda to study token taxes. We utilize scholarly evidence to highlight the core economic risks of AI, situate our proposal for token taxes within the existing literature on robot taxes, and outline a 3-stage audit pipeline for enforcement. We present a research roadmap building off recent work in AI safety and compute governance which enables the implementation of our audit pipeline and the evaluation of the economic impact of token taxes. Our recommendations allow the technical governance research community to leverage token taxes as a promising tool for mitigating the economic risks of AI.

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
