# OpenReview forum: "Position: Token Taxes Can Mitigate AI's Economic Risks"
_ICML.cc/2026/Position_Paper_Track — ICML 2026 Position Paper Track regular_

### Official Review · Reviewer_tFoF · 2026-03-02

**Significance:** 3
**Argument Clarity:** 2
**Rating:** 4
**Confidence:** 4

**Questions:**

See above.

**Alternative Views Section:**

Yes

**Compliance With Llm Reviewing Policy A Conservative:**

Affirmed.

**Discussion Potential:**

3

**Final Justification:**

See my comment below.

**Paper Summary:**

This position paper argues that the ML research community should prioritize studying token taxes to mitigate AI's economic risks, drawing on the historical precedent of Engels' Pause to warn that AI could stagnate wages and living standards for decades. The paper's two key advantages of token taxes over other robot tax proposals are enforceability via a three-stage audit pipeline (black-box token audits, norm-based tax rates, and white-box audits) and their usage-based nature, which ensures developing nations benefit by taxing where AI is consumed rather than where models are hosted. It concludes with five research recommendations while acknowledging counterarguments around innovation disincentives, Jevons' Paradox, geopolitical vetoes by AI superpowers, and the potential superiority of FLOP-based taxation.

**Position:**

Yes

**Position In Title:**

Yes

**Related Work:**

3

**Strengths And Weaknesses:**

**Strengths:**
- I really appreciate the work for two economic reasons: (I) AI has the potential to lead to a significant jump in labor productivity, improving efficiency of humans by orders of magnitude and potentially requiring fewer people on the job. For countries that suffer from demographic decline (e.g., Japan, many EU countries), this sparks hope to hold up (and potentially increase productivity) in many sectors. (II) Taxes are an instrument to steer consumption, finance public infrastructure, and help keep social safety nets up and running. AI infrastructure requires significant public infrastructure, consumes a lot of energy and water, directly competing with private households.
- Related to (II), I think, taxes could be an instrument to move away from simply producing tokens to producing meaningful and useful tokens (i.e., focusing on what can actually help a task). Recent advances like OpenCode or OpenClaw make use of powerful reasoning models that tend to use a large amount of tokens to solve tasks. It is an open question of how useful the tokens are and whether there is an avenue to save on tokens improving the cost efficiency, and subsequently energy and water efficiency.

**Weaknesses:**
- When arguing for token taxes, do you require AI providers to use the same tokenizers? Token efficiency varies greatly depending on what tokenizer is being used, plus the language you use for instructing the models also has an effect. Isn't there a risk for companies and consumers starting to game the system? A token is not a standardized unit like we have agreed on for the metric or imperial system.
- In section 5, I think some of the threat models are over simplified, given that we can use Trusted Execution on CPUs and GPUs for auditing, and enable CPAs and audit firms to run proper analyses, given tokens become a key measure for productivity. Enforcing policy is typically an interdisciplinary challenge where we have to use technical and human efforts jointly. This discussion falls a bit short, I think.
-  Regarding the alternative views (this is an interesting discussion!), there are various aspects that also complicate FLOP-based taxation, given that model precision has huge effects on the FLOPs used for training and inference.I would have liked to get a better understanding of why taxes are likely to hinder innovation in that regard? While I generally think that the government is not a great investor, having good support of public infrastructure is key to innovation, especially in a discipline that requires energy infrastructure to an extent as AI does.
- Given that both tokens and FLOPs are highly dependent on the compute and model configuration, why not simply adapt energy taxes? Measuring energy is standardized, can help incentivize efficient data centers, and is usually closely monitored by certified power meters.

**Support:**

2

---

> ### Author Rebuttal · Authors · 2026-03-30
>
> We thank the reviewer for their kind comments. Our responses to each of the concerns identified above are as follows:
>
> > When arguing for token taxes, do you require AI providers to use the same tokenizers?
>
> We thank the reviewer for this question. We acknowledge that token hasn't been universally standardized and thus open challenges remain for it to be a perfect unit of taxation. We do not argue in favour of a single, uniform token tax rate but rather individual token tax rates for each model family depending on the tokenizer used. Our 3-stage audit pipeline also guarantees that the auditor will have access to the tokenizer used during inference, and can hence protect against gaming by companies.
>
>  We acknowledge two outstanding issues (as you've also pointed out):
>
> 1. For the same sentence written in a specific language, the number of tokens decoded by tokenizers of different LLMs varies.
> 2. For the same sentence (or a semantic meaning) written in two different languages, the number of tokens decoded by tokenizers of those languages varies.
>
> We are less concerned in 1. From the lens of equilibrium, the role of a token tax is to make token efficiency a new variable to be considered for companies using AI. Companies will achieve a new equilibrium where they favor a model that lies on the Pareto frontier of model capability-token / efficiency trade-off. Precisely, companies don't only want a capable model, but also a model that uses fewer tokens to solve the same question. This is beneficial in many aspects, especially given that the number of generated tokens (i.e., number of LLM forward pass) is proportional to the energy consumed.
>
> For 2., we agree that it remains an open challenge requiring further investigation. Our Recommendation (R2), which proposes continuously updating normative tax rates across model categories, partially mitigates this concern. However, we acknowledge that addressing differences in token efficiency across languages—largely driven by users' language proficiency rather than firm-level policy—is an important and under-explored research direction which we value highly. We will highlight these points in the revision.
>
> We note that the issue is not unique to token-based taxation. It also arises under energy taxes and FLOP taxes, since token efficiency is closely tied to the number of model forward passes, and therefore directly affects both energy consumption and computational usage.
>
> > In section 5, I think some of the threat models are over simplified, given that we can use Trusted Execution on CPUs and GPUs for auditing, and enable CPAs and audit firms to run proper analyses... This discussion falls a bit short, I think.
>
> We appreciate the reviewer’s comments on threat models and agree that policy enforcement is an interdisciplinary challenge. This is why we have directed our recommendations to the emerging technical governance community who are uniquely poised to tackle these challenges. We also note that we have limited our recommendations to technical research areas in line with the ICML call and acknowledge the fact that further policy recommendations (such as having CPAs and audit firms run analyses) will likely need to be developed in other venues such as policy venues. We look forward to engaging with these discussions and building on our work in an interdisciplinary environment.
>
> > Regarding the alternative views (this is an interesting discussion!), there are various aspects that also complicate FLOP-based taxation...especially in a discipline that requires energy infrastructure to an extent as AI does.
>
> We thank the reviewer for their comment and agree that FLOP taxes could increase companies’ incentives to improve the precision-capability tradeoff to create models operating at lower FLOPs (by lower precision). However, our focus in the paper was on token taxes and hence the discussion of FLOP taxes is limited.
>
> > Given that both tokens and FLOPs are highly dependent on the compute and model configuration, why not simply adapt energy taxes? Measuring energy is standardized, can help incentivize efficient data centers, and is usually closely monitored by certified power meters.
>
> We take great interest in the reviewer’s idea to develop an AI energy tax, and note that we are open to hybrid approaches as we mention in A4. We also note that energy taxes would still have to be tied to token-use in order to ensure the tax is usage-based and not firm-based. One of the core appeals of token taxes we identify is their usage-based nature which allows AI to be taxed at the point of consumption and hence to compensate the government of the state in which an AI company displaces labour. By taxing energy at source, we lose this advantage. We believe the idea of tying energy use to token use to determine tax liability could be an interesting area to explore and can inform the precise token tax rate as well as the norm rates set by the independent commission we outline in R2.

---

> > ### Author Rebuttal · Reviewer_tFoF · 2026-04-02
> >
> > Thank you for your detailed rebuttal. Most of my concerns have been addressed.
> > I have one follow-up question regarding the per-model-family taxation: Assuming there is an opportunity to decide on a token tax rate per model family, how would you make sure the outcome of the proposed independent commission is non-discriminatory? What makes a fair evaluation process? Variable tax rates for the same service offer plenty of attack surface for lengthy and expensive law suits (and why should someone who hypothetically invents a more efficient tokenizer incur higher taxes? Wouldn't that stifle innovation?).
> > Also, in practice, I fear, this will be a bureaucratic monster (another challenge that might be worth considering in the alternative views section).
> >
> >
> > **A relevant source:**
> >
> > - Supreme Court Considers Whether Sales Tax Exemptions Can Be Discriminatory Taxes, https://taxfoundation.org/research/all/state/supreme-court-considers-whether-sales-tax-exemptions-can-be-discriminatory-taxes/.

---

### Official Review · Reviewer_a3um · 2026-03-12

**Significance:** 3
**Argument Clarity:** 3
**Rating:** 4
**Confidence:** 3

**Questions:**

If norm-based tax rates are adopted, how should the granularity of model categories be defined? For instance, should GPT-4 and GPT-4-turbo be classified into the same category? How are fine-tuned models handled? Overly coarse categorization leads to tax distortions, while overly fine categorization negates the purpose of norm-based taxation—how do the authors suggest balancing this trade-off?

**Alternative Views Section:**

Yes

**Compliance With Llm Reviewing Policy A Conservative:**

Affirmed.

**Discussion Potential:**

3

**Final Justification:**

The rebuttal addressed my main concerns

**Paper Summary:**

This paper argues that the machine learning research community should prioritize the study of token taxes as a technical governance tool to address the economic risks of artificial intelligence (AI). The authors point out that AI-driven automation threatens to erode government tax bases, lower living standards, and disempower citizens.

**Position:**

Yes

**Position In Title:**

Yes

**Related Work:**

2

**Strengths And Weaknesses:**

Strengths:

- The paper astutely identifies the imbalance in AI safety research, which overemphasizes capability risks while neglecting economic risks. It effectively demonstrates the urgency of researching AI’s economic risks, aligning with the ICML community’s growing concern for the social impacts of AI.

- A four-dimensional risk framework is constructed, covering fiscal crises, declining living standards, the disempowerment of citizens, and global inequality. The paper correspondingly analyzes the mitigation mechanisms of token taxes, forming a complete logical chain.

- The paper draws on an extensive body of literature spanning AI safety (Bostrom, Casper), labor economics (Acemoglu, Autor), public finance (Mirrlees, Korinek & Lockwood), and compute governance (Sastry, Heim), with accurate conceptual positioning across disciplines.

Weaknesses:

- Overly narrow definition of Token: The paper focuses primarily on text tokens in large language models (LLMs), yet AI automation encompasses multimodal systems such as computer vision and robotics, for which the concept of a "Token" remains ill-defined.

- Insufficiently clear distinction from FLOPs: Section 6 acknowledges the potential for hybrid use of token taxes and FLOP taxes but fails to clearly define their respective scopes of application.

**Support:**

3

---

> ### Author Rebuttal · Authors · 2026-03-30
>
> We thank the reviewer for their kind comments. Our responses to the concerns outlined above are as follows:
>
> > Overly narrow definition of Token: The paper focuses primarily on text tokens in large language models (LLMs), yet AI automation encompasses multimodal systems such as computer vision and robotics, for which the concept of a "Token" remains ill-defined.
>
> We thank the reviewer for pointing this out. We agree that this adds complexity and note that we have focused on text-only LLMs as the most feasible application for an initial token tax. Extensions to other modalities such as vision and embodied AI/robotics will require further research, which overlaps with the independent commission we propose is R2. Our vision of the commission would see it maintain different tax rates for different modalities (text tokens, CLIP patches, audio) to account for different types of models. The hybrid approaches recommended in response to A4 could also allow for compute normalised “token units” which can generalise to multimodal systems. We encourage further research into expanding the token tax to other modalities, and look forward to engaging with this work.
>
> > Insufficiently clear distinction from FLOPs: Section 6 acknowledges the potential for hybrid use of token taxes and FLOP taxes but fails to clearly define their respective scopes of application.
>
> We thank the reviewer for this question. Concretely, we argue that token taxes should be applied when the AI provider employs clear token-based billing such as for text-based LLMs, whereas FLOP taxes and hybrid approaches should be used where billing is not as clearly defined (such as for other modalities as the reviewer points out). We will make this explicit in the revision.
>
> > If norm-based tax rates are adopted, how should the granularity of model categories be defined? For instance, should GPT-4 and GPT-4-turbo be classified into the same category? How are fine-tuned models handled? Overly coarse categorization leads to tax distortions, while overly fine categorization negates the purpose of norm-based taxation—how do the authors suggest balancing this trade-off?
>
> We thank the reviewer for this point. This is an open technical governance research question which we hope the ICML community will engage with. We believe agent based modeling (ABM) which we recommended in R3 can be utilized to estimate the relationship between categorization level and tax efficiency, and encourage further research in this area.

---

> > ### Author Rebuttal · Reviewer_a3um · 2026-04-02
> >
> > My concerns have been adequately addressed.

---

### Official Review · Reviewer_zhGq · 2026-03-15

**Significance:** 3
**Argument Clarity:** 3
**Rating:** 4
**Confidence:** 4

**Questions:**

Please address my questions under the three parts of the strengths and weaknesses section above.

**Alternative Views Section:**

Yes

**Compliance With Llm Reviewing Policy A Conservative:**

Affirmed.

**Discussion Potential:**

3

**Final Justification:**

While I remain positive about the paper, the theoretical distinction between who owns the model and who provides access to it make the discussion in the paper less practical.

**Paper Summary:**

This paper advocates for the use of token taxes as a way of mitigating AI's economic impact.

The rationale of the authors for the need of a token tax is that AI will reshape economic activity in multiple ways. For example, gains in productivity may reduce the bargaining power of the workforce in similar fashion to what happened during the first industrial revolution, which in turn may affect government revenue because it tends to come more from taxing individuals than companies.

Their proposed way of taxing is based on the logic that taxing AI as a service (as opposed to as capital) allows it to be taxed where it is used, which in turn reflects on revenue for countries with less or no AI infrastructure. That also allows taxing based on use, which distributes the cost according to the benefit obtained by different users.

The authors consider different mechanisms that may need to be used for proper auditing to avoid strategic tax evasion, outline the topics that would require further study before adequate implementation, and make a case for other perspectives on the topic.

**Position:**

Yes

**Position In Title:**

Yes

**Related Work:**

3

**Strengths And Weaknesses:**

## Strengths

1. The authors present a global perspective of ongoing changes that makes it important to consider how to change tax codes in order to adapt with a new economic reality. Putting that in contrast with the prevailing optimistic perspective about AI is also useful.

2. The discussion of a precedent in the first industrial revolution, described as Engel's Pause, is helpful to put in perspective what may be coming up with the frequent layoffs in the tech sector in recent years. But has something similar happened during the second industrial revolution?

3. The authors present a fair consideration of alternative views, or at least a much more benevolent one than in other position papers that I have reviewed.

## Weaknesses

1. Although I believe the name token taxes does justice to how to think about this form of taxation, I believe that it is more important to focus on the logic of taxing the service to consumers at the point of service. Sticking to a token count creates a new set of problems, as discussed by the authors, including hidden reasoning tokens and the use of multiple tokenizations as a form of minimizing taxes due. To put in different terms: taxes of utilities are not based on units of what is provided, such as gallons of water or Watts of energy, but instead over the price paid. Why not characterize and think about it in more general terms?

2. Although the authors consider the adversarial role that AI companies would play if token taxes are imposed, in other parts their description of what companies would do seem disconnected from practice. First, they seem to assume in their framework of token auditing that we can separate model owners from service providers, whereas in practice it is quite common for companies like Anthropic, Google, and OpenAI to act as both. Second, they assume that norm taxes would be developed and supported by partnering with those same companies, such as Anthropic and Perplexity, for obtaining reliable usage data (Page 6). Do the authors assume that there would be regulation leading to breaking down such companies between model owners and service providers (re:first question)? Do the authors assume that it would be in the best interest of such companies to provide truthful and reliable data that would be later used for taxing them (re:second question)?

3. The authors sometimes present token takes as an alternative to robot taxes, and other times as one of the many types of robot taxes. Although other types of robot taxes are alluded to a few times in the text, they are never explicitly listed and described. That makes it harder to consider how token taxes would compare to any of such alternatives, save for (possibly?) FLOP taxes. Which other types of robot taxes exist?

4. There are many gray areas that should probably be acknowledged, even if they cannot be addressed right away. For example, how to deal with VPNs? How can you tax a company that does not operate in the country of use, other than through a website? What about companies serving themselves across borders? What about consumers with undisclosed location paying in cryptocurrency?

5. Assuming that a new Engel's Pause would last exactly 40 years again (Page 1), is too deterministic to be historically acceptable. Why claim that?

## Other issues

- Many hyphens should be replaced by em-dashes (--- in LaTeX), such as in "first half of 2025 alone - a rate rivaling" (Page 1), "corporation serves customers - a phenomenon which" (Page 2), "be inefficient - a conclusion" (Page 7), before "unemployment" and after "inequality" (Page 8), "hosted models - information which" (Page 8)
- Quotation marks should be opened with `` in LaTeX, since '' is used only for closing them, such as in "automation tax" (Page 2), "Compute North" and "Compute South" (Page 3), "harvest" (Page 3), "white box" (Page 5), "model substitution problem" (Page 5)
- When saying "global North and South" (Page 3), why not capitalize G for consistency?
- Instead of "open-source models" (Page 7), a better term would be "open-weight models"
- Towards the end of the text, many citations are plain text and not a link, indicating that they were cited properly in LaTeX. Many of those are repetitions of Korinek & Lockwood (2025), but there is at least one mention of Sastry et al. (2024)
- Page 7, A2, put period after citations instead of before
- Page 8, A3, first line: replace ";" with ":"
- Page 8, A3, end of first paragraph: the sentence is incomplete. What is missing here?
- Page 8, Section 7, suggested addition of text in "are used[, and] not where models are hosted"

**Support:**

3

---

> ### Author Rebuttal · Authors · 2026-03-30
>
> We thank the reviewer for their kind comments and feedback. We respond to their concerns below:
>
> > Although I believe the name token taxes does justice to how to think about this form of taxation, I believe that it is more important to focus on the logic of taxing the service to consumers at the point of service. Sticking to a token count creates a new set of problems, as discussed by the authors, including hidden reasoning tokens and the use of multiple tokenizations as a form of minimizing taxes due. To put in different terms: taxes of utilities are not based on units of what is provided, such as gallons of water or Watts of energy, but instead over the price paid. Why not characterize and think about it in more general terms?
>
> We appreciate the reviewer’s comment and note that the main appeal of the token tax is its usage-based, auditable nature. If an AI tax was simply applied to the price paid, it would still be necessary to ensure that it is 1) enforceable and 2) usage-based. AI firms are large multi-national corporations (MNCs) which are able to avoid traditional robot taxes like corporation taxes by recording profits in favourable jurisdictions. Our 3-stage auditing pipeline makes token taxes more enforceable. Token taxes also mitigate against global inequality by ensuring that the tax is paid in the user’s country and not in the favourable tax jurisdiction chosen by the MNC.
>
> > Do the authors assume that there would be regulation leading to breaking down such companies between model owners and service providers (re:first question)? Do the authors assume that it would be in the best interest of such companies to provide truthful and reliable data that would be later used for taxing them (re:second question)?
>
> We thank the reviewer for this question and agree that it adds complexity. We have focused primarily on open technical governance questions of interest to the community, and acknowledge the fact that these open policy questions will also need to be answered to facilitate adoption of token tax legislation.
>
> > The authors sometimes present token takes as an alternative to robot taxes, and other times as one of the many types of robot taxes. Although other types of robot taxes are alluded to a few times in the text, they are never explicitly listed and described. That makes it harder to consider how token taxes would compare to any of such alternatives, save for (possibly?) FLOP taxes. Which other types of robot taxes exist?
>
> We define robot taxes as an umbrella term for taxes on automation that substitutes for human labor. Such taxes can take multiple forms. For example, a per-robot tax (Korinek & Lockwood, 2025) imposes periodic charges on the ownership or deployment of automated systems (e.g., humanoid robots or automated assembly lines). A FLOPs tax targets the volume of computational operations required to run automated programs. An energy tax instead taxes the energy consumption associated with such systems. Existing forms of robot taxes include corporation taxes, automation levies, and disallowing corporate tax deductions (these are present in the introduction in lines 68-72).
>
> > There are many gray areas that should probably be acknowledged, even if they cannot be addressed right away. For example, how to deal with VPNs? How can you tax a company that does not operate in the country of use, other than through a website? What about companies serving themselves across borders? What about consumers with undisclosed location paying in cryptocurrency?
>
> We thank the reviewer for pointing out these additional issues, and will engage with them in more detail in the revision.
>
>
> > Assuming that a new Engel's Pause would last exactly 40 years again (Page 1), is too deterministic to be historically acceptable. Why claim that?
>
> We thank the reviewer for this question. We did not intend to claim that a modern Engel's Pause would last exactly 40 years, but rather to illustrate the fact that multi-decade long stagnations in living standards are a possibility as a result of AI - the exact length of the stagnation could be longer or shorter (though we believe it could be longer due to AI's transformative potential).

---

> > ### Author Rebuttal · Reviewer_zhGq · 2026-04-02
> >
> > I recommend that the authors check the language used about robot taxes throughout the paper, so that it does not sound as if token taxes are an alternative to robot taxes in some parts while also understood as a form of robot tax in other parts.
> >
> > Moreover, if the analysis ignores the current practice, such as by assuming a distinction between model owners and access providers, then the need for further refinement must be also acknowledged.

---

### Official Review · Reviewer_Bxe5 · 2026-03-18

**Significance:** 2
**Argument Clarity:** 3
**Rating:** 4
**Confidence:** 3

**Questions:**

1. Can you provide an explicit definition and discussion of robot taxes?
2. Can you elaborate on the significance of the economic model discussed in Sec. 3.1, especially around equation (1)? E.g.:
    - What is the rationale behind eq. (1)? What is $\alpha$? What is $\varepsilon$? What is the significance of the $max$ on the left-hand side?
    - How is the robot tax defined in this setting?
    - How does this economic model relate to the proposed token tax?
3. In the paragraph around lines 193-205 in the right-hand column: can you explicitly describe why robot taxes (a) can't be effectively enforced and (b) do not mitigate worsening global inequality?
4. Line 266, left-hand column: what are the "other forms of robot tax" mentioned here? Why are they not enforceable?
5. Line 358-359, left-hand column: by "hardware-level monitoring" of compute resources, do you mean something along the lines of government or auditing authorities physically monitoring all GPUs in use in their jurisdiction?
6. This question relates to "R4: Solving the problem of self-hosted models" and "A1: Token taxes will disincentivize innovation": under the token tax scheme proposed, would AI companies be taxed whenever/wherever they call their own models for research and development or model training purposes? Does this undermine your argument against A1?

**Alternative Views Section:**

Yes

**Compliance With Llm Reviewing Policy A Conservative:**

Affirmed.

**Discussion Potential:**

3

**Final Justification:**

Though some concerns remain, the author rebuttal provided adequate clarification.

**Paper Summary:**

The position of this paper is that the use of AI models should be taxed at the token level, i.e., when and where input is passed into a model. It is argued that this method of taxation is superior to alternative methods (such as robot taxes or FLOP taxes) in that it allows governments to address four main economic dangers from AI: potential tax revenue crises, a repeat of Engel's Pause following the first industrial revolution, human disempowerment, and exacerbated global inequality. A token tax collection scheme is outlined and a roadmap of research directions founded on the goal of enabling such token taxes is proposed. Alternative approaches and objections to the proposed token tax scheme are discussed.

**Position:**

Yes

**Position In Title:**

Yes

**Related Work:**

2

**Strengths And Weaknesses:**

**Strengths**

The token tax scheme described is interesting and a reasonable candidate for addressing the four main AI economic risks outlined in the summary above. The collection scheme is interesting and the three levels of auditing provide research depth, giving rise to several open research directions. The research agenda proposed is substantial, well-reasoned, and appropriate and useful connections to the existing literature that can serve as jumping-off points for future work are provided. Overall, the token tax idea and proposed research directions are likely of interest to the community and may stimulate useful follow-on work.

**Weaknesses**

The core weakness of the paper is that existing tax models -- especially the robot tax, which the paper presents as its main competitor tax model -- are insufficiently discussed. In particular, the robot tax model is not adequately described, making it difficult for the reader to accurately assess the advantages and disadvantages of the proposed token tax model compared with the robot tax model. The economic model of Sec. 3.1 (especially equation (1)) is not discussed in sufficient depth for the reader to understand how the token tax relates to the existing literature.

**Support:**

3

---

> ### Author Rebuttal · Authors · 2026-03-30
>
> We thank the reviewer for their kind comments and feedback. In response to the questions raised above:
>
> > Can you provide an explicit definition and discussion of robot taxes?
>
> We define "robot taxes" as an umbrella term for taxes on automation that substitutes for human labor. Such taxes can take multiple forms. For example, a per-robot tax (Korinek & Lockwood, 2025) imposes periodic charges on the ownership or deployment of automated systems (e.g., humanoid robots or automated assembly lines). A FLOPs tax targets the volume of computational operations required to run automated programs. An energy tax instead taxes the energy consumption associated with such systems. A token tax is a specific instance of a robot tax. It focuses on the volume of tokens processed by language models, which constitute the foundation of advanced AI systems capable of performing tasks traditionally carried out by human labor.
>
> > Can you elaborate on the significance of the economic model discussed in Sec. 3.1, especially around equation (1)?
>
> > What is the rationale behind eq. (1)? What is $\alpha$? What is $\varepsilon$? What is the significance of the $max$ on the left-hand side?
>
> This equation provides a closed-form expression for the maximum attainable government revenue from labor taxation, normalized by total economic output (GDP). It is Proposition 3 in Korinek & Lockwood.
>
> Eq. (1) is derived under the assumption that GDP (Y) follows the Cobb-Douglas production function:
> $Y = A*K^{\alpha}*L^{1-\alpha}$,
> where $\alpha$ denotes production’s share relative to capital stock, A is Total Factor Productivity, K is the Capital Stock, and L is Labor Supply. The parameter $\alpha$ directly enters Eq. (1).
>
> $L$ is further modeled as:
> $L = L_{0}(1 - \tau_{L})^\epsilon$, where $L_{0}$ is a baseline labor supply, $\tau_{L}$ is the labor tax rate, and $\epsilon$ is the elasticity of labor’s with respect to respect to the net-of-tax rate. The $\epsilon$ also propagates into Eq. (1). For full derivation, see Section A.1.4 (pages 40-41) of Korinek & Lockwood.
>
> > How does this economic model relate to the proposed token tax?
>
> Eq. (1) implies that, **as AI substitutes for human labor—i.e., as the capital share $\alpha \to 1$—the maximum achievable revenue from labor taxation converges to zero, regardless of the tax rate $\tau_{L}$**. Additionally, the maximum revenue from labor tax is negatively proportional to $\epsilon$—for a given \tau_{L}, a lower $\epsilon$ leads to a larger labor supply $L$, i.e., a larger tax base of labor tax.
>
> This result supports our argument that widespread automation may erode labor tax and create fiscal pressure, thereby motivating the consideration of alternative instruments such as the token tax.
>
> > In the paragraph around lines 193-205 in the right-hand column: can you explicitly describe why robot taxes (a) can't be effectively enforced and (b) do not mitigate worsening global inequality?
>
> Current robot taxes include existing taxes on technology companies such as corporation taxes and automation levies. In response to (a), robot taxes are not enforceable because AI firms are large multi-national corporations (MNCs) which are able to avoid traditional robot taxes like corporation taxes by recording profits in favourable tax jurisdictions (We explain this in further detail in section 2.1 (lines 80-93)). In contrast, we argue that token taxes are enforceable using the 3-stage auditing pipeline we propose in section 4. In response to (b), robot taxes do not mitigate against worsening global inequality because (i) insufficient tax is collected due to (a) and (ii) robot taxes such as corporation tax can be paid in the jurisdiction chosen by the MNC (and not the jurisdiction in which the MNC raises revenue from). Token taxes mitigate against global inequality by ensuring that the tax is paid in the user’s country and not in the favourable tax jurisdiction chosen by the MNC (i.e token taxes are usage-based).
>
> > Line 266, left-hand column: what are the "other forms of robot tax" mentioned here? Why are they not enforceable?
>
> Alternative forms of robot tax include corporation tax, automation levies, and disallowing corporate tax deductions. These are mentioned in the introduction in lines 68-72. We address the enforceability concern above.
>
> > Line 358-359, left-hand column: by "hardware-level monitoring" of compute resources, do you mean something along the lines of government or auditing authorities physically monitoring all GPUs in use in their jurisdiction?
>
> Yes.
>
> > This question relates to "R4: Solving the problem of self-hosted models" and "A1: Token taxes will disincentivize innovation": under the token tax scheme proposed, would AI companies be taxed whenever/wherever they call their own models for research and development or model training purposes? Does this undermine your argument against A1?
>
> No, token taxes will only be applied as consumption taxes (where a consumer utilizes an AI model).

---

> > ### Author Rebuttal · Reviewer_Bxe5 · 2026-04-03
> >
> > Thanks to the authors for their responses. While I appreciate the clarifications regarding the questions raised, significant additional discussion will need to be added in the paper to fully address them. I maintain my score.

---

### Decision · Program_Chairs · 2026-04-30

**Decision:**

Accept (regular)

**Comment:**

This paper discusses potential economic risks of AI, and token taxes as a potential solution. The paper argues that the advantage of this approach is due to its enforceability (via black-box token audits, norm-based tax rates, and white-box audits) and their usage-based nature, which ensures taxation where AI is consumed rather than where models are hosted. The paper is well-written and may lead to interesting discussion.

The paper targets the technical governance research community, which does not have a large intersection with the ICML community.

In the revision, the authors should add a clear definition of 'robot taxes' and discuss additional issues / alternative views:
- assuming a distinction between model owners and access providers is unrealistic
- tokens are too narrow and ignore multimodal systems
- tokens are not a standard unit - their count will depend on the tokenizer, language, etc.
- bureaucratic hurdles.